# Arginine Methylation Regulates Ribosome CAR Function

**DOI:** 10.3390/ijms22031335

**Published:** 2021-01-29

**Authors:** Kristen Scopino, Carol Dalgarno, Clara Nachmanoff, Daniel Krizanc, Kelly M. Thayer, Michael P. Weir

**Affiliations:** 1Department of Biology, Wesleyan University, Middletown, CT 06459, USA; kscopino@wesleyan.edu (K.S.); cdalgarno@wesleyan.edu (C.D.); cnachmannoff@wesleyan.edu (C.N.); 2Department of Mathematics and Computer Science, Wesleyan University, Middletown, CT 06459, USA; dkrizanc@wesleyan.edu (D.K.); kthayer@wesleyan.edu (K.M.T.); 3College of Integrative Sciences, Wesleyan University, Middletown, CT 06459, USA; 4Department of Chemistry, Wesleyan University, Middletown, CT 06459, USA

**Keywords:** ribosome translocation, molecular dynamics, mRNA GCN periodicity, A-site decoding center, codon adjacency, arginine methylation, stress regulation

## Abstract

The ribosome CAR interaction surface is hypothesized to provide a layer of translation regulation through hydrogen-bonding to the +1 mRNA codon that is next to enter the ribosome A site during translocation. The CAR surface consists of three residues, 16S/18S rRNA C1054, A1196 (*E. coli* 16S numbering), and R146 of yeast ribosomal protein Rps3. R146 can be methylated by the Sfm1 methyltransferase which is downregulated in stressed cells. Through molecular dynamics analysis, we show here that methylation of R146 compromises the integrity of CAR by reducing the cation-pi stacking of the R146 guanidinium group with A1196, leading to reduced CAR hydrogen-bonding with the +1 codon. We propose that ribosomes assembled under stressed conditions have unmethylated R146, resulting in elevated CAR/+1 codon interactions, which tunes translation levels in response to the altered cellular context.

## 1. Introduction

Protein translation is regulated under different cellular conditions by multiple mechanisms including control of translation initiation, elongation, termination, and ribosome biogenesis. Many of these control points are modulated by post-translational modifications such as: through stress-signaled phosphorylation of the alpha-subunit of EIF2 which blocks translation initiation [1,2]; stress-induced inhibitory phosphorylation of EEF2 leading to inhibition of translation elongation [3]; stress-regulated halting of translation through K63 polyubiquitination of ribosome proteins at multiple mainly solvent-exposed sites [4,5]; and stress-mediated modulation of tRNA modifications including nucleotides of the anticodon loop which are important in translation fidelity, frameshifting, and translation efficiency [6,7,8]. Methylation of rRNA or mRNA nucleotides can also modulate translation initiation or elongation [9,10,11]. In a recent study [12], we identified the CAR ribosome surface, which has sequence-sensitive interactions with the mRNA as it enters the ribosome decoding center during translation elongation. In this study, we elucidate a mechanism to modulate CAR function based on cellular conditions.

During protein translation, threading of mRNA through the ribosome is thought to depend on mRNA–ribosome interactions [13,14,15,16]. Modulation of these interactions provides potential pathways for fine-tuning rates of protein translation in different cellular conditions. Cryo-EM studies in yeast have identified five intermediate stages of ribosome translocation (stages I through V; [17]) and we have identified a ribosome interaction surface, named CAR [12,18], that has pronounced hydrogen-bonding to the mRNA during the early translocation stages I and II. CAR interacts with the +1 codon next in line to enter the ribosome A site. This interaction is sequence sensitive with a preference for GCN codons which are enriched in the initial codons of highly-expressed genes. The CAR surface consists of three residues—16S/18S rRNA C1054 and A1196 (*E. coli* numbering), and ribosomal protein Rps3 R146 (*S. cerevisiae* numbering)—and integrity of CAR depends on pi stacking between C1054 and A1196, and cation-pi stacking between A1196 and the guanidinium group of R146. CAR is anchored adjacent to the A-site tRNA anticodon through base stacking between C1054 and tRNA nucleotide 34 (the anticodon partner of the wobble nucleotide; Figure 1). The CAR surface has been studied in translocation ribosome structures of yeast, but to conform to prokaryote studies, we use *E coli* numbering for C1054 and A1196 (*S. cerevisiae* 18S rRNA C1274, and A1427, respectively).

While characterizing the CAR interaction surface, our attention was drawn to the observation that R146 can be modified to ω-N^G^-monomethylarginine by the yeast Sfm1 methyltransferase [19]. Although no ortholog of Sfm1 has yet been found in higher eukaryotes, Rps3 R146 is highly conserved across eukaryotes, and human S3 can be methylated by yeast Sfm1 [20]. Ribosome profiling experiments have revealed that expression of Sfm1 is downregulated under various stress conditions (oxidative stress, heat shock, starvation; Appendix A; [21]). These observations led us to ask whether down regulation of R146 methylation under stress conditions might impact the CAR–mRNA interaction. We speculated that addition of a methyl group to the R146 guanidinium group might interfere with its stacking with A1196 and disturb the integrity of the CAR surface. Our molecular dynamics analysis presented here confirmed that methylation of R146 disrupts CAR integrity leading to reduced H-bonding between CAR and the +1 codon, providing a potential mechanism for tuning protein translation levels.

## 2. Results

### 2.1. Ribosome Subsystems Provide Neighborhoods for Molecular Dynamics

Molecular dynamics (MD) studies have provided important insights into ribosome function [22,23]. This has included the use of ribosome subsystems to characterize important steps in recognition of the correct tRNA through interactions of 16S/18S A1492, A1493 and G530 (*E. coli* numbering) with the minor groove of the codon-anticodon base pairs in the A-site decoding center [22,24,25]. In a previous study [12], we characterized interactions of the CAR surface with the +1 codon next to enter the A site. We used a subsystem of the yeast ribosome with 495 residues—183 rRNA nucleotides and 312 ribosomal protein amino acids—and applied a restraining force on an “onion shell” of residues around the surface of the subsystem in order to maintain the topologies and structures of the different translocation stages of the yeast ribosome identified in cryo-EM studies [12].

The subsystem used in our published study, henceforth referred to as neighborhood 1 (N1), was centered around G530 of the 530 loop, located adjacent to the A-site decoding center. In order to provide a wider region between the onion shell and our residues of interest (the CAR interface, +1 codon, and A-site codon and anticodon), we decided to design a second subsystem, neighborhood 2 (N2), centered around C1054 of the CAR interaction surface. N2 consists of 494 residues—252 nucleotides and 242 amino acids—with 173 residues in the onion shell and 321 unrestrained residues (Appendix A; Appendix A).

To compare neighborhoods 1 and 2, we performed 30 independently-initiated MD replicates in N2 (20 × 60-ns runs, and 10 X 100-ns runs, for a total of 2.2 μs) using translocation stage II which in N1 showed strong interactions between CAR and the +1 codon. Levels of H-bonding between CAR and the +1 codon were slightly higher in N2 compared to N1 (Appendix A). A key aspect of CAR function is that the interaction between CAR and the +1 codon is sensitive to the sequence at the +1 codon. Using N1, we showed previously that GCU at the +1 codon leads to strong CAR/+1 codon interactions, whereas replacement of the second nucleotide (GGU, GAU, or GUU) led to significantly reduced interactions [12]. The same relationship was observed with N2 (Appendix A) confirming our previous conclusion that the CAR interaction is sequence sensitive with a preference for GCN codons, and suggesting that sensitivity to translation regulation through the CAR interface likely depends on the degree to which codons in a gene’s open reading frame (ORF) conform to GCN. In this paper, we report results using N2. Similar assessments showing equivalent results for N1 are provided in Appendix A.

### 2.2. R146 Methylation Disrupts CAR Integrity

Integrity of the CAR interface likely depends on geometric rigidity conferred by stacking interactions between C1054, A1196, and the guanidinium group of R146 [26,27,28]. The R146 amino acid can be methylated at NH1 or NH2 of the arginine guanidinium group (PDB naming) [19,20]. We hypothesized that the R146 methylation could disrupt cation-pi stacking of the guanidinium group with the A1196 base. To test this, we introduced a methyl group at NH2 and performed molecular dynamics using the translocation stage II, subsystem (20 × 60-ns runs, and 10 × 100-ns runs).

Root mean square deviation (RMSD) assessment was performed using either the structure at the beginning of production dynamics, or the average structure across the full trajectory, as the reference for the RMSD calculations. Both methods showed that equilibrium dynamics were reached by 20 ns of dynamics for both the methylated R146 (mR146) and unmethylated (R146) trajectories (Appendix A). Hence, we used trajectories after 20 ns for all assessments for a total of 1.6 µs each for methylated and unmethylated experiments.

Root mean square fluctuation (RMSF) assessment was performed for CAR and nearby residues—the CAR, +1 codon, and A-site codon, and anticodon residues—to determine whether methylation of R146 affected fluctuations of the arginine or other residues. The RMSF analysis was restricted to the heavy atoms of the base rings for RNA nucleotides and the guanidinium group for R146 (excluding the methyl group in mR146; Figure 2 legend). This revealed that R146 has significantly elevated fluctuations in mR146 compared to R146 (*t*-test *p* < 0.001; Figure 2), suggesting that methylation of R146 might affect stacking of the guanidinium group with A1196.

To assess stacking more directly, we measured the distances between the center of mass of the A1196 base rings and the guanidinium group center of mass. The center-of-mass distance was higher for mR146 compared to R146 (Figure 3A), supporting the conclusion that methylation led to reduced cation-pi stacking.

To further test this conclusion, we used a modified version of the solvent accessible surface area (SASA; [29,30]) calculation to measure the van der Waals molecular surface area accessible to a water-sized probe. Limited accessibility between the A1196 base and (m)R146 guanidinium group provided an indication of stacking between these residues. To focus the SASA measurement on A1196 and (m)R146, trajectories were stripped of all other residues, and to ensure equivalent measurements in the R146 and mR146 systems, we created stripped versions that only retained the four heavy atoms of the guanidinium group (without the methyl modification) and the nine heavy atoms of A1196. These “stripped SASA” comparisons showed that mR146 had significantly more accessible surface compared to R146 (*t*-test *p* < 0.01; Figure 3B), indicating decreased stacking when arginine is methylated.

### 2.3. R146 Methylation Reduces CAR/+1 Codon Hydrogen Bonding

Since methylation of R146 reduces CAR integrity, we hypothesized that the CAR/+1 codon interaction might also be compromised. We measured overall hydrogen bonding between CAR and the +1 codon for the mR146 and R146 runs. The first (G1) and second (C2) nucleotides of the +1 codon showed higher levels of hydrogen bonding with the CAR residues for R146 compared to mR146 (Figure 4A,B). This difference was significant for C2 (*t*-test *p* < 0.001; Figure 4A). Equivalent results were obtained using the N1 subsystem (Appendix A). To confirm that the reduced hydrogen bonding was due to methylation of R146, the methyl group was removed from the last frames of the mR146 trajectories (20 × 60-ns trajectories) and MD was restarted (after energy minimization, heating, and equilibration steps). Hydrogen bonding between CAR and the +1 codon C2 was partially restored in these demethylated structures (Appendix A) suggesting that the reduced hydrogen bonding of mR146 was a result of methylation. Hence, in addition to disruption of CAR integrity, methylation of R146 also leads to reduced CAR/+1 codon interaction, potentially providing the basis for a mechanism to tune translation under different cellular conditions.

### 2.4. Cross-Trajectory Comparisons Show Variability of Behavior

Focusing on the 12 residues of the CAR/+1 codon and A-site codon/anticodon, we compared behaviors across multiple trajectories initiated with independent assignments of atom velocities. RMS2D [31] pairwise comparisons of 820 frames drawn from 20 independent trajectories (1 frame/ns from 20 to 60 ns) were performed. When backbone atoms were used, mR146 trajectories showed slightly lower variability between experiments than R146 (Figure 5A, right triangles below leading diagonals of heatmaps). However, RMS2D using the “core” heavy atoms from base rings and the R146 guanidinium group showed higher variability between mR146 experiments compared to R146 (Figure 5A, upper left triangles). This difference was slightly less pronounced when (m)R146 was excluded from the RMS2D measurements (Figure 5B), consistent with the observation that the methylated R146 guanidinium group has elevated RMSF (Figure 2). However, the difference between R146 and mR146 was still significant when (m)R146 was excluded (bootstrap analysis *p* < 0.01; Figure 5B) suggesting that the methylation difference at R146 changed the dynamics of other residues in the neighborhood. Similar results were obtained (Appendix A) using 810 frames from 10 longer trajectories for mR146 and R146 (1 frame/ns from 20 to 100 ns).

The RMS2D assessments (Figure 5A, Appendix A) also revealed greater levels of variability within each trajectory for mR146 compared to R146. This higher granularity within the heat map blocks for each experiment was particularly pronounced for the mR146/R146 comparison for core atoms of the bases and guanidinium group (Figure 5A upper left triangles). These observations are consistent with the greater fluctuation of the guanidinium group in mR146 detected in RMSF assessment (Figure 2).

Trajectories were also inspected using the VMD visualization program [32,33]. For each trajectory, key observations of the A-site codon/anticodon residues and CAR/+1 codon residues were recorded (Appendix A; Videos S1 and S2). In general, methylated R146 was observed to be more dynamic than unmethylated R146, with less stacking to A1196 and less H-bonding interaction to the +1 codon. In addition, nucleotide 2 of the +1 codon was oriented towards the CAR interface more consistently with R146, potentially facilitating codon-tRNA interaction as translocation proceeds (Appendix A). Representative examples of trajectories are shown in Videos S1 and S2.

## 3. Discussion

### 3.1. A Key Role for Arginine Methylation

This study points to a key role for Rps3 R146 in the interaction of the ribosome CAR surface with mRNA. It is estimated that arginine is involved in about half of RNA-protein H-bond interactions and arginine methylation by several methyltransferases is the most common post-translational modification of ribosomal proteins (e.g., detected in HeLa cells; [34]). Stacking of the arginine guanidinium group commonly contributes to RNA-protein interactions [34]. Although methylation of the guanidinium group does not change pKa appreciably [35], the methylation is reported to have subtle effects such as reducing H-bonding and increasing bulk, stacking interactions, and hydrophobic interactions [34]. In our present study, we observed that methylation of R146 led to reduced H-bonding to the +1 codon but decreased stacking with its A1196 partner in the CAR interface. 

Sfm1 mono-methylates R146 of Rps3 [19,20] and this methyltransferase appears to target one other (unidentified) yeast protein [36]. Sfm1 is unique among SPOUT methyltransferases, in that it is unlikely to methylate rRNA due to a negatively charged surface near its binding site [20]. An *SFM1* (*YOR021C*) ortholog has not yet been found in humans, but yeast Sfm1 can mono- and di-methylate human S3 in vitro [20]. Interestingly, *SFM1* (*YOR021C*) deletion mutants do not have a lethal phenotype in yeast, suggesting that R146 methylation may be responsible for more fine-grained control of translation regulation [19,37]. Because R146 methylation is stoichiometric [19] and no demethylases have yet been found [38], it is likely that R146 is methylated during ribosome biogenesis and that, as discussed below, ribosomes with unmethylated R146 are produced during turnover under stress conditions.

R146 is conserved across eukaryotes, but not prokaryotes [12,18]. Mutational analysis has shown that when R146 is replaced in yeast Rps3, this can lead to lethal or slow growth phenotypes depending on the amino acid substitution, and R146 substitution can affect translation initiation and fidelity [39,40]. These observations, together with our assessment of the effects of R146 methylation, suggest that R146 has a functional role in translation that is likely mediated by the CAR interface.

### 3.2. Rps3 R146 Methylation Modulates CAR Function

This study demonstrates that methylation of R146 of the Rps3 protein modulates CAR function (Figure 6). When R146 is unmethylated, CAR has strong H-bond interactions with the +1 codon during ribosome translocation. However, when R146 is methylated, CAR integrity is disrupted and the CAR/mRNA interactions are significantly weakened. We hypothesize that different cell conditions lead to switching between assembly of ribosomes with methylated or unmethylated R146. The Sfm1 methyltransferase responsible for methylating R146 in yeast is downregulated under various stress conditions (Appendix A; [21]) suggesting that ribosomes assembled during stress conditions have unmethylated R146 leading to strong CAR/+1 codon interactions.

Since the CAR/mRNA interaction is highly sensitive to the sequence of the +1 codon, with a strong preference of GCN codons [12,18], the sensitivity of different mRNAs to regulation by CAR will depend upon its codon content. The initial codons of ORFs, the “ramp” region, appear to be particularly influential in modulating overall translation levels of mRNAs [41], perhaps analogous to an entry ramp to a highway determining traffic flow. Indeed, protein-coding ORFs tend to have a subtle three-nucleotide GCN periodicity [42,43], and many genes with high protein expression are particularly rich in GCN codons in their ramp regions suggesting that these genes have high sensitivity to CAR-mediated regulation [12,18].

In this study, we have focused on Rps3 R146, one of the three CAR residues. A variety of nucleotide modifications are detected in rRNA and tRNA residues, and modifications are common in the neighborhood of the A-site decoding center [9,44]. While many of these ribosome and tRNA modifications appear to be constitutive, some of the modifications may be regulated by different cell conditions [9,45,46] and we hypothesize that some modifications in addition to R146 methylation might also contribute to modulating the CAR behavior. Indeed, it is possible that integration of additional modifications may enhance the magnitude of the change in CAR/mRNA hydrogen bonding upon methylation of R146.

### 3.3. Conclusions

Gene expression is regulated at many levels including control of translation. There are multiple modes of translation regulation and we have investigated here a layer of regulation that is sensitive to mRNA codon sequences and hypothesized to be modulated depending on cell conditions. Under stress conditions such as heat, anoxia, or starvation, newly assembled ribosomes are postulated to have an unmethylated R146 in their CAR interaction surface leading to high levels of hydrogen bonding to the +1 codons about to enter the A site during ribosome translocation. Increasing the CAR/+1 codon interaction is thought to act like an accelerator or brake depending upon the protein expression levels of a gene and the degree of hydrogen bonding. Expression studies [18] suggest that increasing CAR/+1 interactions in the ramp regions of lower expression genes tends to increase their expression whereas the opposite is true for high-expression genes with GCN-rich codons in the ramp regions of their protein coding sequence. It is likely that future studies of CAR/mRNA interactions with different codons in the A-site and +1 positions, and the effects of ribosome modifications on these interactions, will provide further insights into this new layer of translation regulation.

## 4. Materials and Methods

### 4.1. Subsystem Neighborhoods

To simulate and analyze the region of the yeast ribosome containing the decoding center and the CAR interaction surface, we used the Amber18 and AmberTools18 software packages [47]. The initial coordinates were sourced from previously published cryo-EM structures (PDB ID: 5JUP; [17]). To facilitate extensive sampling of trajectories, a subsystem model of the ribosome was used. As discussed in Results, this subsystem referred to as neighborhood 2 (N2) had similar behavior to the neighborhood 1 (N1) used previously [12].

The N2 subsystem was designed based on a 35 Å-radius sphere centered around C1054 of the CAR interaction surface. An important design consideration was to create a contiguous outer shell of residues that would be restrained during dynamics to maintain the structural integrity and specific translocation stage conformation of the subsystem. This “onion shell” layer is approximately 8 Å thick. A small number of residues were added to the initial 35 Å-radius sphere selection in order to reduce the number of artificial chain breaks, holes in the onion shell, and transitions between restrained and unrestrained residues. Neighborhood 2 consists of a total of 494 residues; the outermost 173 residues are restrained and the innermost 321 residues are unrestrained. From this selection of residues, the PDB was modified to remove 5′ phosphates from nucleotide chains and to modify the mRNA sequence from 5′-AAUGCCUGCUAAC-3′ to 5′-AAUGCCUGCUGCC-3′ in order to conform to a GCN periodicity. For both nucleotide replacements, the maximum number of atomic positions were retained so that tLEaP [47] could grow the rest of the atoms in an orientation similar to the resolved cryo-EM structure.

### 4.2. Molecular Dynamics Setup

The ff14SB force field was used for protein and the ff99bsc_chiOL3 force field for RNA [48,49,50]. Parameters for the methylated arginine came from the Forcefield_PTM set [51]. A modified histidine residue (DDE) of eEF2 was parameterized using antechamber. The system was solvated using the TIP3P [52] water model in a 12.0 Å octahedral box with K^+^ ions to achieve electroneutrality [53]. Energy minimization, heating, and equilibration steps following our previously described protocol ([12]; File S1), were carried out prior to replicate dynamics (20 × 60-ns and 10 × 100-ns runs, for a total of 2.2 μs). Of note, the dynamics in Neighborhood 2 were more stable in energy than in Neighborhood 1, allowing us to remove a 50 ps step previously needed for the onion shell residues to evolve before fixing the onion shell atom coordinates used as a reference structure to restrain (at 20 kcal/mol Å^2^) the onion shell through the trajectory.

### 4.3. Trajectory Analysis

The dynamics analysis was conducted using cpptraj functions [54], as previously described [12], with the following changes and additional methods (Data File S1). RMSD values were computed using the average structure for a given experiment as the reference and used to determine the average time for equilibration dynamics to be reached (20 ns for both methylated and unmethylated R146 trajectories). Root mean square fluctuation (RMSF) by residue was calculated using the cpptraj rmsf command, using the average structure as the reference. The RMS2D matrix was calculated by concatenating together experimental trajectories with reduced sampling (1 frame per ns) and using the rms2d cpptraj command. The core atom calculations for RMSF, SASA, and RMS2D included the heavy atoms of the base rings for nucleotides (C2, C4, C5, C6, N1, N3 of C and U; C2, C4, C5, C6, C8, N1, N3, N7, N9 of A and G) and the heavy atoms of the guanidinium group for R146 (CZ, NE, NH1, NH2; the methyl group was not included for mR146). Backbone atom calculations for RMSD, RMSF, and RMS2D included backbone atoms N, CA, C, and carbonyl O for amino acids and sugar-phosphate O5′, O3′, P, and C5′ for nucleotides. Bonferroni corrections were applied to *t*-test results.

The modified solvent accessible surface area (SASA; [29,30]) calculation, optimized to quantitate the extent of cation-pi stacking, involved stripping the trajectories down to only A1196, only R146, and only A1196 and R146 together. These residues were then further stripped to the “core” atom versions. The cpptraj surf command was then used to calculate stripped SASA for each of these trajectories.

## Figures and Tables

**Figure 1 ijms-22-01335-f001:**
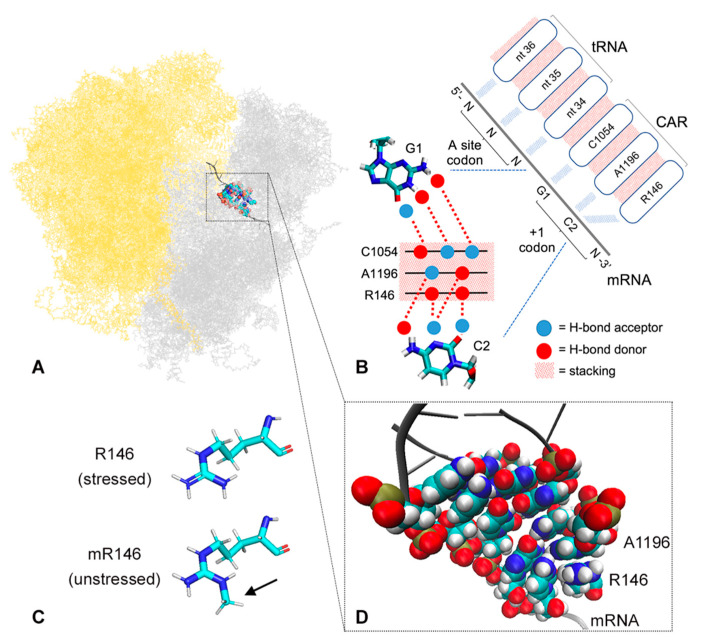
The CAR interaction surface. (**A**) Cryo-EM structure (PDB ID 5JUP; [17]) of translocation stage II yeast ribosome showing large (yellow) and small (grey) subunits. Additionally, highlighted in color are the A site codon/anticodon, CAR interface, and +1 codon next to enter the A site. (**B**) Cartoon showing interaction between the CAR interface and first (G1) and second (C2) nucleotides of the +1 codon. H-bond donors are illustrated in red and H-bond acceptors in blue. (**C**) Rps3 R146 is hypothesized to be ω-N^G^-monomethylated (arrow) in unstressed cells, and unmethylated in stressed cells. (**D**) van der Waal sphere representation of CAR interface showing stacking between the R146 guanidinium group and the A1196 base of the CAR interaction surface. Atoms are colored: N (blue), C (cyan), O (red), H (white), P (olive).

**Figure 2 ijms-22-01335-f002:**
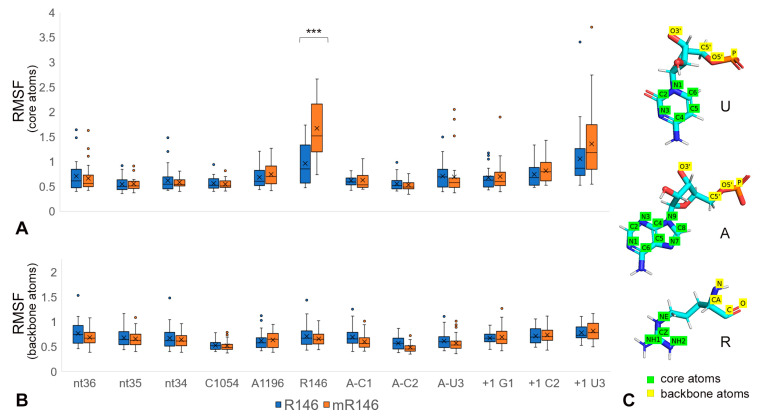
Methylated R146 shows elevated fluctuation. RMSF (Å) was calculated for residues comparing 30 trajectories with methylated (mR146) and unmethylated (R146) arginine. RMSF is shown for residues: CAR (C1054, A1196, (m)R146), +1 codon (+1 G1, +1 C2, +1 U3), A-site codon (A-C1, A-C2, A-U3) and tRNA anticodon (nt36, nt35, nt34). (**A**) shows RMSF calculated with “core” heavy atoms of bases (C2, C4, C5, C6, N1, N3 of C and U; C2, C4, C5, C6, C8, N1, N3, N7, N9 of A and G) and the guanidinium group (CZ, NE, NH1, NH2). mR146 has significantly greater fluctuations than R146 (*t*-test *p* < 0.001 ***). (**B**) shows RMSF for backbone heavy atoms. No significant differences between the mR146 and R146 trajectories were observed in measurements of backbone RMSF. (**C**) Illustration of core and backbone atoms used for RMSF.

**Figure 3 ijms-22-01335-f003:**
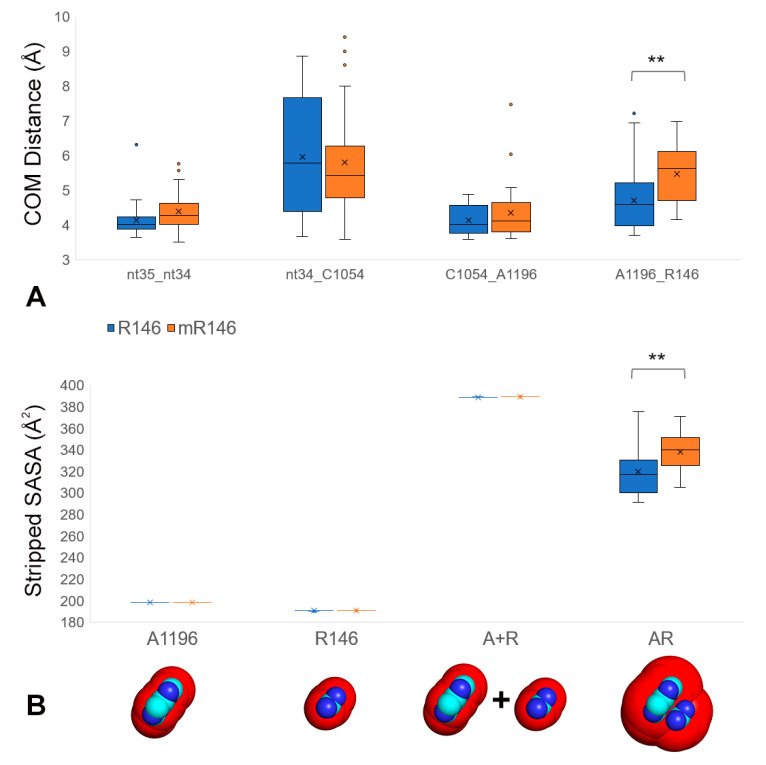
R146 methylation disrupts cation-pi stacking between R146 and A1196 in the CAR interface. (**A**) Stacking was assessed by measuring the distances between the centers of mass of the heavy atoms of bases or the guanidinium group of R146. Center of mass (COM) distances were measured for anticodon nt35 to nt34, anticodon nt34 to C1054, C1054 to A1196, and A1196 to R146, comparing the trajectories for methylated (mR146) and unmethylated (R146) guanidinium group. 30 trajectories were compared for each. The center of mass distance is significantly higher for mR146-A1196 compared to R146-A1196 (*t*-test *p* < 0.01 **). (**B**) A stripped version of the solvent accessible surface area (SASA) protocol (see Materials and Methods) was used to assess solvent accessibility between (m)R146 and A1196 (in Å^2^). The methylated guanidinium group had significantly higher solvent accessibility than unmethylated (*t*-test *p* < 0.01 **), suggesting that the methylation disrupts cation-pi stacking with A1196.

**Figure 4 ijms-22-01335-f004:**
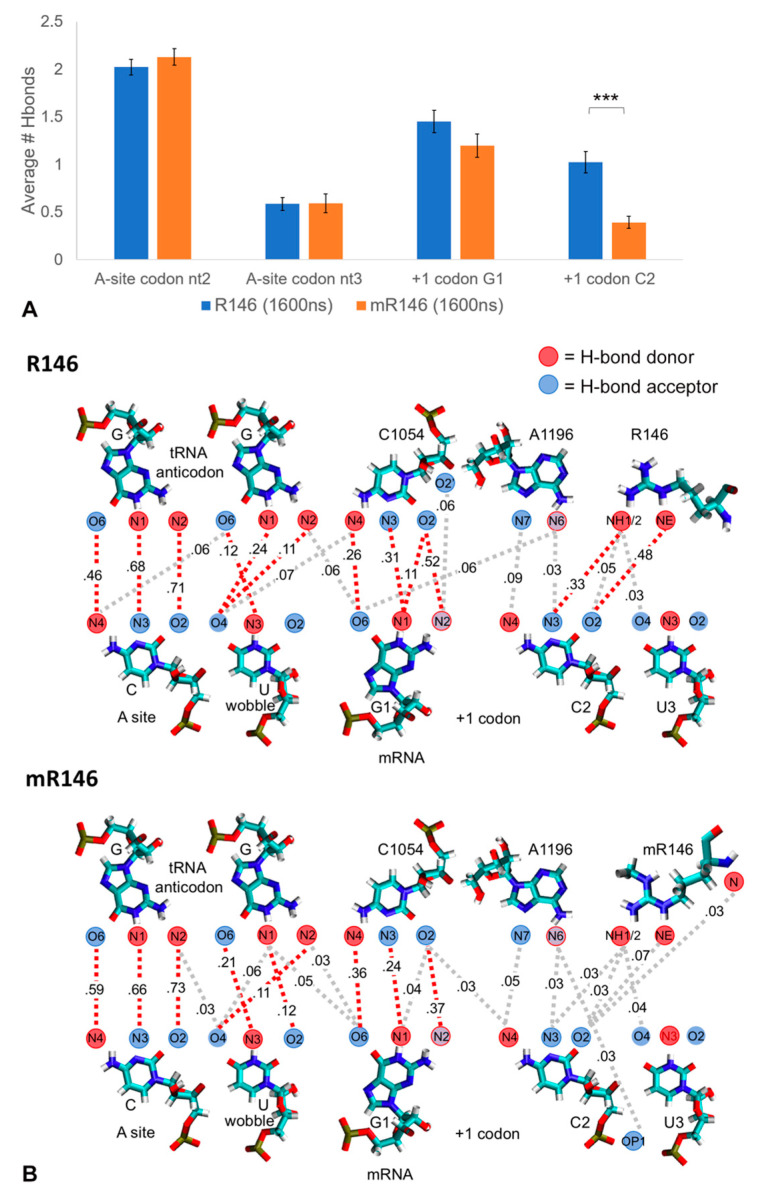
R146 methylation reduces CAR/+1 codon H-bonding. H-bond interactions were quantitated for the A-site nucleotides 2 and 3 and the three +1 codon nucleotides with unmethylated (1.6 µs) and methylated (1.6 µs) R146. All hydrogen bonds were counted between these residues and the anticodon and CAR residues (e.g., interactions of C2 with C1054, A1196 and R146). (**A**) Overall H-bond counts show significantly depressed H-bonds for the second nucleotide of the +1 codon (C2) comparing methylated and unmethylated R146 trajectories (*t*-test *p* < 0.001 ***). (**B**) Illustrated are all H-bond interactions with frequencies *f*: 0.025 < *f* < 0.095 (grey dashes) or *f* > 0.095 (red dashes). This reveals depressed H-bond interactions for methylated R146 compared to unmethylated.

**Figure 5 ijms-22-01335-f005:**
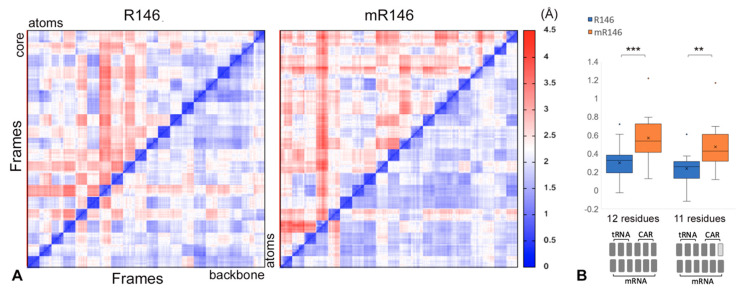
Comparisons across trajectory experiments. (**A**) 20 independently-initiated trajectories were subjected to pairwise RMSD comparisons (between all pairs of frames) using backbone heavy atoms (bottom right-hand triangles below the diagonals) or “core” base and guanidinium group heavy atoms (top left-hand triangles) for 12 residues: CAR (C1054, A1196, (m)R146), +1 codon (+1 G1, +1 C2, +1 U3), A site (A-C1, A-C2, A-U3) and tRNA anticodon (nt36, nt35, nt34). The methylated R146 subsystem had a greater difference between RMSD of backbone and core atoms (right) compared to unmethylated R146 (left). Each trajectory was sampled with 41 frames (1 frame per ns). (**B**) For each of the 20 trajectories, the average RMSD difference with the other 19 trajectories was calculated. The differences (in Å) between the core and backbone atom RMSD means (averaged by experiment) were graphed. Bootstrap analysis indicated that mR146 trajectories had a significantly greater difference than for R146 (12 residue; *p* < 0.001 ***; 10,000 bootstrap samples). This was also true when the R146 residue was excluded from the RMS2D calculations (11 residue; *p* < 0.01 **).

**Figure 6 ijms-22-01335-f006:**
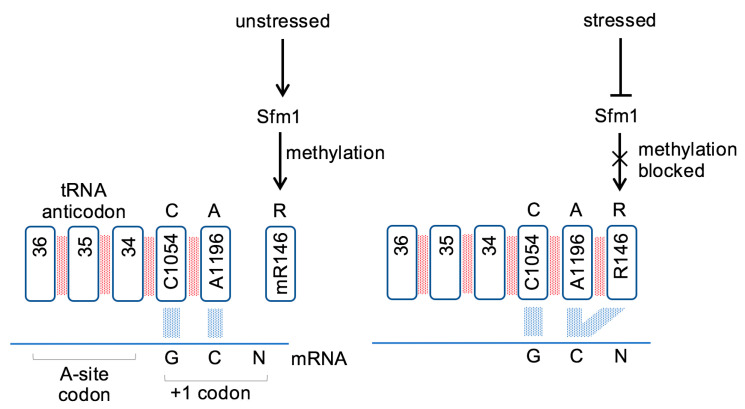
Arginine methylation regulates ribosome CAR function. Methylation of Rps3 R146 is hypothesized to downregulate CAR function by reducing cation-pi stacking of the R146 guanidinium group with A1196 which in turn reduces H-bonding of CAR with the +1 codon next to enter the ribosome A site. Expression of Sfm1, the methyltransferase responsible for R146 methylation, is down regulated in response to stress, promoting CAR integrity and CAR/+1 codon H-bonding.

## Data Availability

The molecular dynamics trajectory data presented in this study are available on request from the corresponding author. The data are not publicly available due to very large file sizes.

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
