# Peer review of "Arginine Methylation Regulates Ribosome CAR Function"

_ijms, 2021, doi:10.3390/ijms22031335_

Round 1

Reviewer 1 Report

In "Arginine Methylation Regulates Ribosome CAR Function 2", the authors perform simulation within and with methylation at R146 and find that methylation disrupts the structure and dynamics of the CAR site, as well as its interaction with the +1 codon. While I think the paper could use some experimental validation and probably could have been merged with the author's last paper on CAR function, it passes the technical bar and I believe just requires one additional set of simulations.

The authors perform many short replicates, which I believe was a good idea as it ensures that their results are robust to thermal noise. However, it is not clear that the observed destabilization is due to methylation - it could be due to the fact that the system was perturbed in general. If the destabilization was due to methylation, one would expect that if you took the destabilized mR146 systems (say, the last frames of the trajectories), demethylated them, and then simulated them again, they would re-stabilize in the original R146 state. If they do not re-stabilize, it really can not be said that the observed behavior comes from the methylation in particular. I believe the authors should perform this computational experiment to verify that the destabilization is in fact due to methylation.

From the figures alone, I count 38 tested hypotheses. If one were to do a conservative multiple hypothesis correction, I believe the only tests that would survive would be the p < 0.001 results. This doesn't change the conclusions much qualitatively, but I think the authors should use MHC in some way. 

Reviewer 2 Report

See attached

Round 2

Reviewer 1 Report

I thank the authors for their response to my concerns and believe the paper is acceptable for publication.